# Research on the relationship between physical literacy and demographic variables and interpersonal support for physical exercise among adolescents in China

Weisong Chen[1], Bowei Zhou[2], Xuan Wang[1]*, Lin Li[1]

1 Sports Training Academy, Chengdu Sport University, Chengdu, Sichuan, China, 2 College of Physical Education, Southwestern University of Finance and Economics, Chengdu, Sichuan, China

* xuanwang0415@163.com

## Abstract

In recent decades, the concept of "Physical Literacy" has emerged as a significant breakthrough in enhancing the physical health of children and adolescents. This concept has become the guiding principle and future direction for physical education reform in Western countries. As China's economic and social development enters a new phase, the negative impact of insufficient physical activity on adolescent health remains a significant challenge to the strategies of "Healthy China" and "A Leading Sports Nation." The unique value of physical literacy, which emphasizes the importance of regular physical activity, aligns perfectly with the developmental needs of the nation, the era, and individuals. Given the crucial role of interpersonal support in the development of adolescents' physical literacy, this study employs questionnaire surveys and statistical analysis to investigate the current state of physical literacy among Chinese adolescents. It explores the predictive relationship between parental support, teacher support, and peer support on physical literacy, aiming to provide guidance for the future development of adolescents' physical literacy. The research findings indicate significant differences in the five dimensions of physical literacy (attitude, sports knowledge, emotion, athletic ability, and physical fitness) among adolescents of different grades and residence type. Additionally, there are notable gender differences in the dimensions of sports knowledge, emotion, and athletic ability. The interaction effect of gender and grade is particularly evident in the emotional and physical fitness dimensions. There is a significant correlation between interpersonal support and all dimensions of physical literacy. After controlling for variables such as gender, grade, and family background, interpersonal support for adolescent physical exercise was found to have a significant impact on all dimensions of physical literacy. Among the forms of support, peer support was identified as a more powerful predictor of physical literacy compared to parental and teacher support.

**Data Availability Statement:** All relevant data are within the manuscript and its Supporting Information files.

**Funding:** The author(s) received no specific funding for this work.

**Competing interests:** The authors have declared that no competing interests exist.

# 1. Introduction

Physical literacy is an essential component for promoting lifelong physical activity and health. Recognized globally, it encompasses not only physical fitness but also the motivation, confidence, and knowledge to engage in physical activities throughout life [1]. Physical literacy has been adopted by many Western countries as a cornerstone for reforming physical education to address the health challenges posed by sedentary lifestyles [2].

In China, the concept of physical literacy has gained significant attention [3], particularly following the State Council's issuance of the "Outline of Building a Sports Powerful Country" in September 2019. This outline emphasizes the improvement of physical literacy and health across the population, setting ambitious goals: by 2020, significant improvements in physical literacy and health; by 2035, a substantial enhancement in the physical literacy and health status of young people; and by 2050, achieving world-leading levels in these areas. The integration of sports and education is seen as a crucial strategy for fostering lifelong physical activity and health, contributing to the grand goal of national rejuvenation and modernization [4].

Despite the strategic emphasis on physical literacy, modern lifestyles characterized by technological advancements and convenience have led to a decline in daily physical activity, especially among adolescents [5]. This sedentary trend poses significant health risks, including the rise of chronic diseases and the deterioration of physical health among young people [6]. Current initiatives to promote physical activity have yielded limited success, often due to a lack of motivation and conceptual understanding of physical activity's importance [7].

This study aims to investigate the current state of physical literacy among Chinese adolescents and examine the role of interpersonal support from parents, teachers, and peers in enhancing physical literacy. By utilizing a comprehensive survey and statistical analysis, this research seeks to:

1. Identify key demographic factors influencing physical literacy.

2. Explore the predictive relationship between various forms of interpersonal support and physical literacy development among adolescents.

Understanding these dynamics provides actionable insights for policymakers and educators, directly contributing to the strategic goals of "Healthy China" and "A Leading Sports Nation." Specifically, the findings can inform the design of targeted interventions that foster supportive environments at home and in schools. Promoting a culture of active participation in physical activities aligns with the objectives of "Healthy China," improving the overall physical fitness and health of the population. Moreover, enhancing physical literacy among youth supports the long-term goal of establishing China as a leading sports nation by 2050, ensuring a healthier, more physically active generation equipped with the knowledge, skills, and motivation to maintain lifelong physical activity.

In conclusion, this study not only reinforces the importance of physical literacy as a national priority but also provides a detailed roadmap for leveraging interpersonal support to achieve broader health and sports development goals. This makes it a critical piece of research in the ongoing efforts to cultivate a healthier, more active, and physically literate youth population in China.

# 2. Theoretical framework

## 2.1 Existing theoretical models

**2.1.1 Social ecological model (SEM).** The Social Ecological Model (SEM) emphasizes the multiple levels of influence on individual behavior, including intrapersonal, interpersonal,

organizational, community, and policy levels [8]. This model recognizes that behaviors are affected by interactions between individuals and their physical and social environments [9]. In the context of youth physical literacy, SEM helps to understand how various environmental factors, such as family, school, community, and broader societal influences [10], contribute to or hinder the development of physical literacy.

Research has shown that various factors affect the physical fitness of children and adolescents, including genetic factors, morphological development, physical exercise, learning and living habits [11]. Key acquired factors include heavy academic burdens, insufficient physical exercise, sedentary lifestyles, and lack of exercise motivation [12]. Changes in lifestyle, academic pressures, school and social sports environments, parental attitudes, and the status of physical education are also significant influences [13].

Participation in physical exercise is known to improve physical health. Studies based on SEM discuss the influencing factors of physical exercise behavior at individual, family, school, community, and policy levels [14]. Important factors include motivation to participate in sports, family health awareness, economic and educational environment, professional quality of PE teachers, school exercise environment, and community facilities [15]. The formation of adolescents' exercise habits is influenced by individuals, families, schools, and society [16], with school-level factors such as teachers' professional quality, teaching philosophy, and methods being particularly significant [17].

**2.1.2 Self-Determination theory (SDT).** Self-Determination Theory (SDT) focuses on the intrinsic and extrinsic motivations driving human behavior. It posits that fulfilling the basic psychological needs for autonomy, competence, and relatedness is essential for fostering intrinsic motivation and well-being [18]. SDT provides a framework for understanding how supportive social environments can enhance motivation for physical activities [19], which is crucial for developing physical literacy.

Interpersonal support means that individuals feel the support of significant others for their free choices and independent decisions [20], and can obtain valuable information, emotional recognition, and reduced pressure from significant others [21]. According to SDT, people tend to engage in activities according to their values and interests, but their motivation and behavior are also influenced by the social environment [22]. When the social environment provides autonomy support, individuals are more likely to fully tap their internal resources and actively adapt and develop under external encouragement [23]. Empirical studies have found that autonomy support significantly predicts academic and mental health outcomes in adolescents [24,25].

**2.1.3 Empirical evidence on interpersonal support.** Empirical research indicates that parental support is a crucial factor influencing children's participation in physical activities [26]. The interpersonal support environment in schools, including support from teachers and peers, also significantly impacts adolescents' physical exercise [27]. Family is an essential domain for cultivating youth sports cognition and establishing good habits [28]. Parental support positively influences children's engagement in physical exercise, boosting their decision-making confidence and self-control, thus stabilizing their exercise behaviors [29].

In the school context, teacher support provides a foundation for learning, imitating, and standardizing behaviors, helping students establish correct sports values and healthy lifestyles [30]. Peer support offers external motivation for maintaining exercise behaviors, improving social adaptability, relieving stress and loneliness, and establishing exercise habits through shared experiences [31]. Studies have shown that the more interpersonal support individuals receive, the easier it is to establish a sense of value, belonging, and active engagement in physical activities [32–34]. Interpersonal support becomes an important "cue" to internalize and stimulate individual behaviors, indirectly promoting the occurrence and maintenance of exercise behaviors [35].

## 2.2 Theoretical innovations of this study

**2.2.1 Quantifying interpersonal support.** This study uniquely quantifies the specific contributions of parental, teacher, and peer support to physical literacy. By using a comprehensive survey and statistical analysis, we provide empirical evidence on how different forms of interpersonal support predict various dimensions of physical literacy. This approach fills a gap in the existing literature by providing a detailed understanding of the impact of each type of support.

**2.2.2 Contextual adaptation.** Considering the unique socio-cultural context of China, this study adapts and extends existing theoretical models to fit the Chinese educational and social environment. It addresses specific challenges and opportunities within this context, providing tailored recommendations for policy and practice to promote physical literacy among Chinese youth. By doing so, it offers insights into how cultural and societal factors influence the effectiveness of interpersonal support in promoting physical activity.

## 2.3 Conceptual model

The core focus of this study is to examine the impact of interpersonal support on the physical literacy of adolescents. Interpersonal support, encompassing parental support, teacher support, and peer support, plays a critical role in shaping the physical activity behaviors and attitudes of young individuals.

In this conceptual model, we posit that interpersonal support significantly influences the development of physical literacy in the following ways:

**Parental support.** Parents influence their children's physical activity by providing encouragement, resources, and modeling active behaviors. Parental attitudes towards physical activity can shape their children's perceptions and motivations, leading to higher levels of engagement and physical literacy.

**Teacher support.** Teachers, particularly physical education instructors, are pivotal in delivering quality physical education programs and fostering a positive attitude towards physical activity. Their support can enhance students' skills, knowledge, and confidence in engaging in physical activities.

**Peer support.** Peers provide social reinforcement and a sense of belonging, which are important for adolescents. Positive peer interactions and support can motivate adolescents to participate in physical activities, thereby enhancing their physical literacy.

This model integrates these three sources of interpersonal support to provide a comprehensive understanding of their collective impact on adolescents' physical literacy. By employing a comprehensive survey and statistical analysis, this study aims to quantify the specific contributions of each type of support and understand their interactions.

Our research hypothesizes that higher levels of parental, teacher, and peer support are positively associated with higher levels of physical literacy among adolescents. The findings from this study will offer empirical evidence to inform the development of targeted interventions that leverage these sources of support to enhance physical literacy.

## 3. Materials and methodology

### 3.1 Participants and data

**3.1.1 Sample size justification.** To ensure the robustness and reliability of our study, we conducted a thorough sample size estimation using G*Power software and general sociological research principles.

We performed a power analysis using G\*Power 3.1 to determine the appropriate sample size for our multiple regression analysis, which includes three independent variables (peer support, teacher support, and parental support). Assuming a medium effect size ($f^2 = 0.15$), a significance level ($\alpha$) of 0.05, and a desired power (1-$\beta$) of 0.80, the analysis indicated that a minimum sample size of 77 participants would be required to detect statistically significant effects.

In addition to the G\*Power analysis, we followed general principles commonly accepted in sociological research, which recommend that the number of participants should be at least 10 to 15 times the number of items in the questionnaire [36]. Our study utilized a physical literacy scale with 42 items and an interpersonal support scale with 24 items, totaling 66 items. According to this guideline, the required minimum sample size would range from 660 to 990 participants (66 items \* 10 participants to 66 items \* 15 participants).

By combining these two approaches, we aimed to ensure our sample size was more than adequate. Our final sample comprised 2,969 adolescents, far exceeding both the G\*Power minimum requirement and the empirical guideline threshold. This large sample size not only enhances the statistical power of our study but also improves the generalizability and reliability of our findings across different demographic groups.

**3.1.2 Participant selection process.**　Participants were selected using a stratified random sampling technique to ensure representation across different regions and demographic groups, thereby minimizing potential selection bias. The sampling frame included primary and secondary schools in grades 5–9 across 15 provinces and cities in China.

The randomization process was applied at multiple levels: first, regions were stratified and schools were randomly selected within these strata. Then, within each selected school, students were randomly chosen to participate. This multi-level randomization ensured a representative sample of the diverse adolescent population across China. The selection process involved the following detailed steps:

**Stratification by region.**　China was divided into five geographical strata based on socio-economic and cultural diversity. Three provinces or cities were selected from each region to ensure balanced representation. The specific regions and selected provinces are shown in Table 1.

**Random selection of schools.**　Within each geographical stratum, schools were selected using a random number generator. The number of schools selected from each region was proportional to the student population in that region, ensuring balanced representation. For example, if a region had 20% of the total student population, 20% of the schools were selected from that region.

**Random selection of students.**　Within each selected school, students from grades 5 to 9 were chosen using a random number generator. A list of students in each grade was obtained, and a random number generator was used to select the required number of students from these lists. This ensured that every student had an equal chance of being selected, further reducing selection bias.

**Table 1. Stratification by region and province in the sampling frame.**

| Region | Province |
|---|---|
| Eastern China | Shanghai, Jiangsu, Shandong |
| Central China | Henan, Hubei, Hunan |
| Western China | Sichuan, Guizhou, Chongqing |
| Southern China | Guangdong, Guangxi, Hainan |
| Northern China | Beijing, Shaanxi, Liaoning |

**Exclusion of certain regions.**   Due to logistical constraints, geographical remoteness, and difficulties in obtaining research permissions, certain regions such as Xinjiang, Tibet, and Taiwan were not included in the sampling frame. Xinjiang and Tibet present unique challenges due to their vast geographical areas, remote locations, and diverse ethnic compositions, which complicate travel and data collection logistics. Taiwan poses additional challenges in obtaining research permissions and ensuring participant cooperation. While their exclusion may limit the generalizability of the findings to these specific regions, the included regions still provide a broad and representative sample of the Chinese adolescent population.

### 3.1.3 Data collection methods

Data collection was conducted using standardized procedures to ensure the validity and reliability of the data. The process included the following steps:

**Training of data collectors.**   Prior to the administration of questionnaires, P.E. teachers and head teachers received comprehensive training. This training covered the objectives of the study, the importance of random sampling, and standardized instructions for administering the questionnaires to ensure consistency.

**Questionnaire administration.**   The questionnaires were administered in a controlled environment during regular school hours. P.E. teachers and head teachers oversaw the process to ensure that students completed the questionnaires independently and without any external influence. The completed questionnaires were collected on the spot to ensure immediate and accurate data recovery. The survey was conducted from September 1, 2022 to December 1, 2022.

**Confidentiality and consent.**   The study adhered strictly to the Declaration of Helsinki and relevant national and institutional guidelines. Ethical approval was obtained from the Ethics Committee of Chengdu Sport University. Given that all participants were minors, verbal consent was obtained from each participant prior to their participation. Additionally, we obtained consent from the parents or guardians of the minors involved in the study. In cases where it was not feasible to obtain parental consent directly, the research ethics committee specifically waived the need for their consent, ensuring compliance with ethical standards. Data analysis was conducted anonymously to maintain confidentiality and protect participants' privacy.

**3.1.4 Data handling.**   A total of 3,300 questionnaires were distributed, with 2,969 valid responses obtained, yielding an effective recovery rate of 89.97%. Invalid questionnaires were excluded based on predefined criteria, including incomplete responses and inconsistent answers, effectively managing missing data by excluding them from the analysis. This approach ensured the quality and integrity of the data. Basic information of respondents is shown in Table 2.

## 3.2 Measurement

Physical literacy was assessed using a scale developed by Xuan Wang [33], which consists of 42 items across five dimensions: emotion, attitude, physical fitness, sports knowledge, and athletic ability. This five-factor structure explains 68.072% of the cumulative variance. Responses are recorded on a 7-point Likert scale ranging from 1 (strongly disagree) to 7 (strongly agree). The scale has demonstrated robust reliability and validity in prior studies.

Parental, teacher, and peer support were measured using scales revised by Liu Jiajing [21]. Parental support was assessed with 12 items, for example, "My parents encourage me to express my ideas and opinions about participating in physical exercise." Teacher and peer support were measured using modified versions of the Perceived Autonomy Support subscale

**Table 2. Basic information of respondents.**

| Variables | Option | Number of people | Percentage |
|---|---|---|---|
| Gender | Male | 1530 | 51.5% |
| | Female | 1439 | 48.5% |
| Grade | Primary school-5 | 447 | 15.1% |
| | Primary school-6 | 310 | 10.4% |
| | Junior high school -7 | 477 | 16.1% |
| | Junior high school-8 | 696 | 23.4% |
| | Junior high school-9 | 1039 | 35.0% |
| Family address | Cities | 1325 | 44.6% |
| | Towns | 740 | 24.9% |
| | Villages | 904 | 30.4% |

from the Exercise Atmosphere scale, each consisting of 6 items. Examples include "My physical education teacher encourages me to ask questions about my participation in physical exercise" for teacher support, and "My peers encourage me to speak up about my participation in physical exercise" for peer support. All responses were recorded on a 5-point Likert scale ranging from 1 (strongly disagree) to 5 (strongly agree).

The indicators of the scales used in the study are summarized in Table 3.

### 3.3 Statistical analysis

**3.3.1 Descriptive statistics.** Descriptive statistics were computed for the basic information of valid samples and the scores of each scale. These statistics provide an overview of the data, including means, standard deviations, and the proportion of scores relative to the maximum possible score for each scale. This helps in understanding the general trends and distributions within the dataset.

**3.3.2 Reliability analysis.** Reliability analysis was conducted to test the stability and consistency of the measurement tools used in the study. The internal consistency of the scales was assessed using Cronbach's alpha. A higher Cronbach's alpha indicates greater reliability and lower standard error, suggesting that the scale consistently measures the intended constructs. All scales showed good internal consistency, with Cronbach's alpha coefficients exceeding 0.7.

**3.3.3 Analysis of variance (ANOVA).** ANOVA was used to explore the differences in physical literacy dimensions among adolescents with different demographic variables, including gender, grade, residence type, and the interaction between gender and grade. The use of ANOVA allows for the identification of significant differences between groups. Post-hoc tests were conducted to further explore these differences.

**Table 3. Indicators of the scales used in the study.**

| Scales | | Number of Items | Scoring Method | Source |
|---|---|---|---|---|
| Physical Literacy (PL) | Emotion | 12 | Likert 7-point (1–7) | Xuan Wang(2022) |
| | Attitude | 11 | Likert 7-point (1–7) | |
| | Physical Fitness | 10 | Likert 7-point (1–7) | |
| | Sports Knowledge | 4 | Likert 7-point (1–7) | |
| | Athletic Ability | 5 | Likert 7-point (1–7) | |
| Parental Support (PAS) | | 12 | Likert 5-point (1–5) | Liu Jiajing (2021) |
| Teacher Support (TES) | | 6 | Likert 5-point (1–5) | |
| Peer Support (PES) | | 6 | Likert 5-point (1–5) | |

Formulas:

$$F = \frac{Between-group\ variance}{Within-group\ variance} \qquad (1)$$

**3.3.4 Correlation analysis and multiple regression analysis.** Correlation analysis was performed to examine the relationships between physical literacy dimensions and interpersonal support dimensions (parents, teachers, peers). Partial correlation analysis was used to control for the effects of demographic variables (gender, grade, residence type).

Formulas:

$$r = \frac{\sum(X_i - \bar{X})(Y_i - \bar{Y})}{\sqrt{\sum(X_i - \bar{X})^2 \sum(Y_i - \bar{Y})^2}} \qquad (2)$$

Multiple regression analysis was conducted to determine the predictive effects of interpersonal support on physical literacy. This method helps in understanding how much of the variance in physical literacy can be explained by interpersonal support after accounting for demographic variables.

Formulas:

$$Y = \beta_0 + \beta_1 X_1 + \beta_2 X_2 + \beta_3 X_3 + \beta_4 C_1 + \beta_5 C_2 + \beta_6 C_3 + \varepsilon \qquad (3)$$

Where Y is the dependent variable (physical literacy), $X_1$, $X_2$, $X_3$ are independent variables (interpersonal support dimensions), $C_1$, $C_2$, $C_3$ are control variables (gender, grade, residence type), and $\varepsilon$ is the error term.

**3.3.5 Handling missing data.** Missing data were managed by excluding invalid questionnaires based on predefined criteria, such as incomplete responses and inconsistent answers. This method effectively manages missing data by ensuring that only valid responses are included in the analysis, thereby maintaining the quality and integrity of the dataset.

**3.3.6 Addressing potential confounding variables.** The analysis controlled for potential confounding variables by including demographic variables (gender, grade, residence type) as covariates in the multiple regression models. This helps to isolate the effects of interpersonal support on physical literacy, ensuring that the observed relationships are not confounded by these demographic factors.

## 4.Result

### 4.1 Descriptive statistics and reliability analysis

**4.1.1 Descriptive statistics of interpersonal support for physical exercise.** Table 4 presents the descriptive statistics of interpersonal support (parents, teachers, and peers) based on the sample data. The parental support scale (PAS) had a score range of 18 to 60, with a mean score of 47.35 (SD = 8.08), representing 78.92% of the total possible score. The teacher support scale (TES) had a score range of 6 to 30, with a mean score of 23.97 (SD = 4.74), representing

**Table 4. Descriptive statistics of interpersonal support for physical exercise.**

| Category | Score range | M | SD | Proportion of mean in total score |
|----------|-------------|-------|------|-----------------------------------|
| PAS | 18–60 | 47.35 | 8.08 | 78.92% |
| TES | 6–30 | 23.97 | 4.74 | 79.90% |
| PES | 6–30 | 23.25 | 4.86 | 77.50% |

**Table 5. Descriptive statistics of physical literacy.**

| Category | Score range | M | SD | Proportion of mean in total score |
|---|---|---|---|---|
| Attitude | 11–77 | 61.88 | 12.74 | 80.36% |
| Sports knowledge | 4–28 | 20.63 | 5.49 | 73.68% |
| Emotion | 12–84 | 65.62 | 16.12 | 78.12% |
| Athletic ability | 5–35 | 27.53 | 6.70 | 78.66% |
| Physical fitness | 10–70 | 53.18 | 11.82 | 75.97% |

79.90% of the total possible score. The peer support scale (PES) also had a score range of 6 to 30, with a mean score of 23.25 (SD = 4.86), representing 77.50% of the total possible score. According to the proportion of the mean score in the total score of each scale, the level of teacher support was relatively higher.

**4.1.2 Descriptive statistics of adolescents' physical literacy.** Table 5 presents the descriptive statistics of physical literacy, which contains 5 dimensions and a total of 42 items, utilizing a 7-point Likert scoring method. The attitude dimension included 11 items, with an average score of 61.88 (SD = 12.74), representing 80.36% of the total possible score. The sports knowledge dimension included 4 items, with an average score of 20.63 (SD = 5.49), representing 73.68% of the total possible score. The emotion dimension included 12 items, with an average score of 65.62 (SD = 16.12), representing 78.12% of the total possible score. The athletic ability dimension included 5 items, with an average score of 27.53 (SD = 6.70), representing 78.66% of the total possible score. The physical fitness dimension included 10 items, with an average score of 53.18 (SD = 11.82), representing 75.97% of the total possible score. According to the proportion of the average score in the total score for each dimension, the attitude dimension had the highest score level, while the sports knowledge dimension had the lowest score level.

**4.1.3 The reliability analysis.** Table 6 presents the results of the internal consistency reliability analysis of the sample data. The Cronbach's α coefficients for the interpersonal support scales (parents, teachers, and peers) were all higher than 0.7, indicating good internal consistency reliability. Specifically, the Cronbach's α coefficients were 0.779 for parental support (PAS), 0.734 for teacher support (TES), and 0.749 for peer support (PES). For the adolescent physical literacy scale, except for the dimension of sports knowledge, which had a Cronbach's α coefficient of 0.768, all other dimensions had coefficients greater than 0.85, indicating good internal consistency for each dimension.

## 4.2 Differences in demographic variables of adolescent physical literacy

**4.2.1 Gender and grade differences in adolescent physical literacy.** This section explores the influence of gender, grade, and the interaction effect of gender and grade on adolescent

**Table 6. Internal consistency reliability of interpersonal support and physical literacy.**

| Category | Item | Cronbach's α |
|---|---|---|
| PAS | 12 | 0.779 |
| TES | 6 | 0.734 |
| PES | 6 | 0.749 |
| Attitude | 11 | 0.915 |
| Sports knowledge | 4 | 0.768 |
| Emotion | 12 | 0.954 |
| Athletic ability | 5 | 0.893 |
| Physical fitness | 10 | 0.915 |

**Table 7. Multivariate ANOVA overall test of gender and grade on physical literacy.**

|  | Value | F | Hypothesis df | P |
|---|---|---|---|---|
| Intercept | .962 | 14871.667 | 5.000 | .000 |
| Gender | .010 | 5.779 | 5.000 | .000 |
| Grade | .160 | 24.621 | 20.000 | .000 |
| Gender * Grade | .012 | 1.748 | 20.000 | .020 |

Note: Only Pillai's Trace tests are reported.

physical literacy. The five dimensions of adolescent physical literacy were taken as the dependent variables, and the gender and grade of adolescents were taken as the grouping variables.

The overall test results are shown in Table 7. The results indicate that the main effects of gender and grade are significant, and there is a significant interaction effect between gender and grade on adolescent physical literacy.

The overall significant effect was analyzed using one-way ANOVA. The results showed significant differences in sports knowledge, emotion, and athletic ability based on gender variables (F = 7.898, $P < 0.01$; F = 15.627, $P < 0.001$; F = 5.742, $P < 0.05$). Significant differences were also found in the five dimensions of physical literacy among adolescents of different grades. The interaction effect between gender and grade was evident in the two dimensions of emotion and physical fitness (F = 3.604, $P < 0.01$; F = 2.508, $P < 0.05$). The results are shown in Table 8.

The significant main effects and interaction effects were analyzed, and the results showed that: In terms of gender variables, there are significant differences in the three dimensions of sports knowledge, emotion and athletic ability, which is mainly reflected in that boys have higher levels than girls in these three dimensions.

Post-hoc pairwise comparisons of each dimension of adolescents' physical literacy by grade revealed that the levels of all dimensions, except for sports knowledge, were significantly higher in grade 6 adolescents compared to other grades. There were significant differences in all dimensions of physical literacy between grade 5 and grades 7 to 9, showing that grade 5 students scored higher in all dimensions of physical literacy compared to those in grades 7 to 9. Grade 7 adolescents exhibited significantly lower levels in all dimensions of physical literacy compared to other grades. There were no significant differences in the dimensions of physical literacy between grade 8 and grade 9 adolescents. Specific results are shown in Tables 9 and 10.

On the emotion dimension (Fig 1), the interaction effect between gender and grade shows that boys across all grades exhibit higher emotion levels than girls of the same grade. Specifically, grade 6 boys have the highest emotion levels, while grade 7 girls have the lowest.

In the dimension of physical fitness (Fig 2), the interaction effect between gender and grade shows that boys in grades 6, 8, and 9 have higher physical fitness levels than girls in the same grades, while girls in grades 5 and 7 have higher physical fitness levels than boys in the same grades. The boys in grade 6 exhibit the highest physical fitness levels, whereas the boys in grade 7 exhibit the lowest.

**4.2.2 Residence type differences in adolescent physical literacy.** Based on the level of economic development and infrastructure, the residence type is categorized into three groups: city, town, and village. Using the five dimensions of adolescent physical literacy as dependent variables and residence type as the grouping variable, the multivariate analysis of variance results (Table 11) indicate that the overall test is significant.

A one-way analysis of variance was conducted to examine the differences in physical literacy dimensions based on residence type. The results (Table 12) showed that there were significant differences in each dimension of physical literacy across different residence types.

**Table 8. One-way ANOVA of gender and grade on physical literacy.**

| Difference sources | Dependent variable | SQ | df | MS | F |
|---|---|---|---|---|---|
| Gender | Attitude | 228.254 | 1 | 228.254 | 1.542 |
| | Sports knowledge | 227.020 | 1 | 227.020 | 7.898** |
| | Emotion | 3683.150 | 1 | 3683.150 | 15.627*** |
| | Athletic ability | 238.101 | 1 | 238.101 | 5.742* |
| | Physical fitness | 272.124 | 1 | 272.124 | 2.197 |
| Grade | Attitude | 41742.036 | 4 | 10435.509 | 70.513*** |
| | Sports knowledge | 3900.429 | 4 | 975.107 | 33.926*** |
| | Emotion | 62104.334 | 4 | 15526.084 | 65.873*** |
| | Athletic ability | 9506.007 | 4 | 2376.502 | 57.310*** |
| | Physical fitness | 44569.553 | 4 | 11142.388 | 89.956*** |
| Gender * Grade | Attitude | 618.487 | 4 | 154.622 | 1.045 |
| | Sports knowledge | 110.893 | 4 | 27.723 | .965 |
| | Emotion | 3398.259 | 4 | 849.565 | 3.604** |
| | Athletic ability | 367.326 | 4 | 91.832 | 2.215 |
| | Physical fitness | 1242.769 | 4 | 310.692 | 2.508* |
| Error | Attitude | 437917.669 | 2959 | 147.995 | |
| | Sports knowledge | 85048.128 | 2959 | 28.742 | |
| | Emotion | 697428.081 | 2959 | 235.697 | |
| | Athletic ability | 122702.759 | 2959 | 41.468 | |
| | Physical fitness | 366515.363 | 2959 | 123.865 | |
| Total | Attitude | 11849161.000 | 2969 | | |
| | Sports knowledge | 1353375.000 | 2969 | | |
| | Emotion | 13555799.000 | 2969 | | |
| | Athletic ability | 2382899.000 | 2969 | | |
| | Physical fitness | 8810159.000 | 2969 | | |

Note

*$P<0.05$

**$P<0.01$

***$P<0.001$.

Further post-hoc pairwise comparative analysis was conducted on the dimensions of physical literacy among adolescents from different residence types. The results (Tables 13 and 14) indicated that adolescents whose families reside in cities had significantly higher scores in all dimensions of physical literacy compared to those in towns and villages. Specifically, adolescents in cities outperformed their peers in towns and villages across all dimensions. However, there were no significant differences in the physical literacy dimensions between adolescents in towns and villages.

**Table 9. Multiple comparisons of physical literacy among different grades 1 (M ± SD).**

| Category | Grade 5 | Grade 6 | Grade 7 | Grade 8 | Grade 9 |
|---|---|---|---|---|---|
| Attitude | 64.79±.58 | 67.72±.70 | 54.28±.56 | 61.76±.46 | 62.41±.38 |
| Sports knowledge | 21.93±.25 | 20.85±.31 | 18.13±.25 | 20.93±.20 | 20.92±.17 |
| Emotion | 70.59±.73 | 72.59±.89 | 56.86±.70 | 65.42±.58 | 65.46±.48 |
| Athletic ability | 29.07±.31 | 30.38±.37 | 23.98±.30 | 27.71±.24 | 27.48±.20 |
| Physical fitness | 55.58±.53 | 60.41±.64 | 45.74±.51 | 53.14±.42 | 53.37±.35 |

**Table 10. Multiple comparisons of physical literacy among different grades 2 (Mean difference).**

| Grade | | Attitude | Sports knowledge | Emotion | Athletic ability | Physical fitness |
|---|---|---|---|---|---|---|
| Grade 5 | Grade 6 | -3.16** | .95 | -2.51 | -1.5** | -5.19*** |
| | Grade 7 | 10.50*** | 3.80*** | 13.74*** | 5.10*** | 9.79*** |
| | Grade 8 | 3.01*** | 1.00* | 5.17*** | 1.37** | 2.37** |
| | Grade 9 | 2.39** | 1.02** | 5.21*** | 1.61*** | 2.16** |
| Grade 6 | Grade 7 | 13.66*** | 2.85*** | 16.24*** | 6.60*** | 14.98*** |
| | Grade 8 | 6.17*** | .05 | 7.68*** | 2.87*** | 7.56*** |
| | Grade 9 | 5.55*** | .07 | 7.72*** | 3.11*** | 7.36*** |
| Grade 7 | Grade 8 | -7.49*** | -2.81*** | -8.56*** | -3.74*** | -7.42*** |
| | Grade 9 | -8.11*** | -2.78*** | -8.52*** | -3.49*** | -7.63*** |
| Grade 8 | Grade 9 | -.62 | .02 | .04 | 0.25 | -0.21 |

Note

*$P<0.05$

**$P<0.01$

***$P<0.001$.

### 4.3 Research on the relationship between adolescent physical literacy and physical exercise interpersonal support

**4.3.1 Correlation between adolescent physical literacy and physical exercise interpersonal support.** Controlling for the variables of gender, grade, and residence type, a partial correlation analysis was conducted on the relationship between adolescent physical exercise

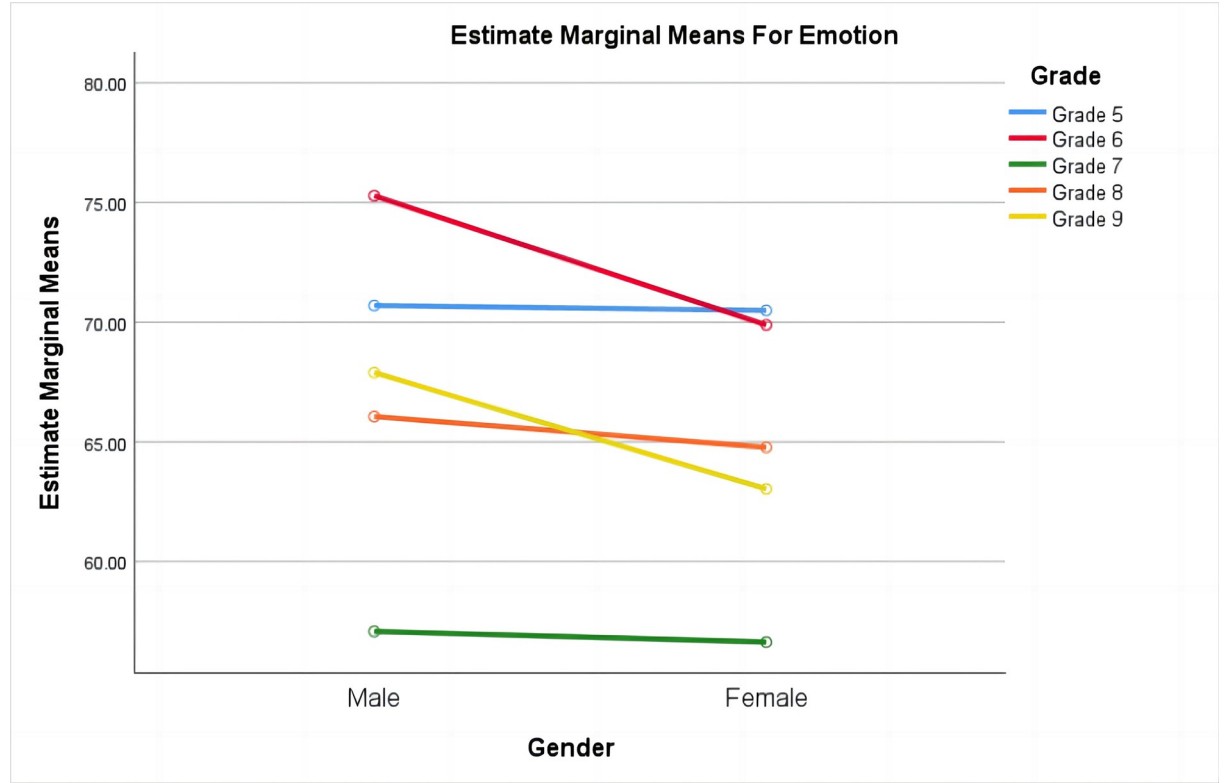

**Fig 1. Gender-grade comparison of emotion dimensions.**

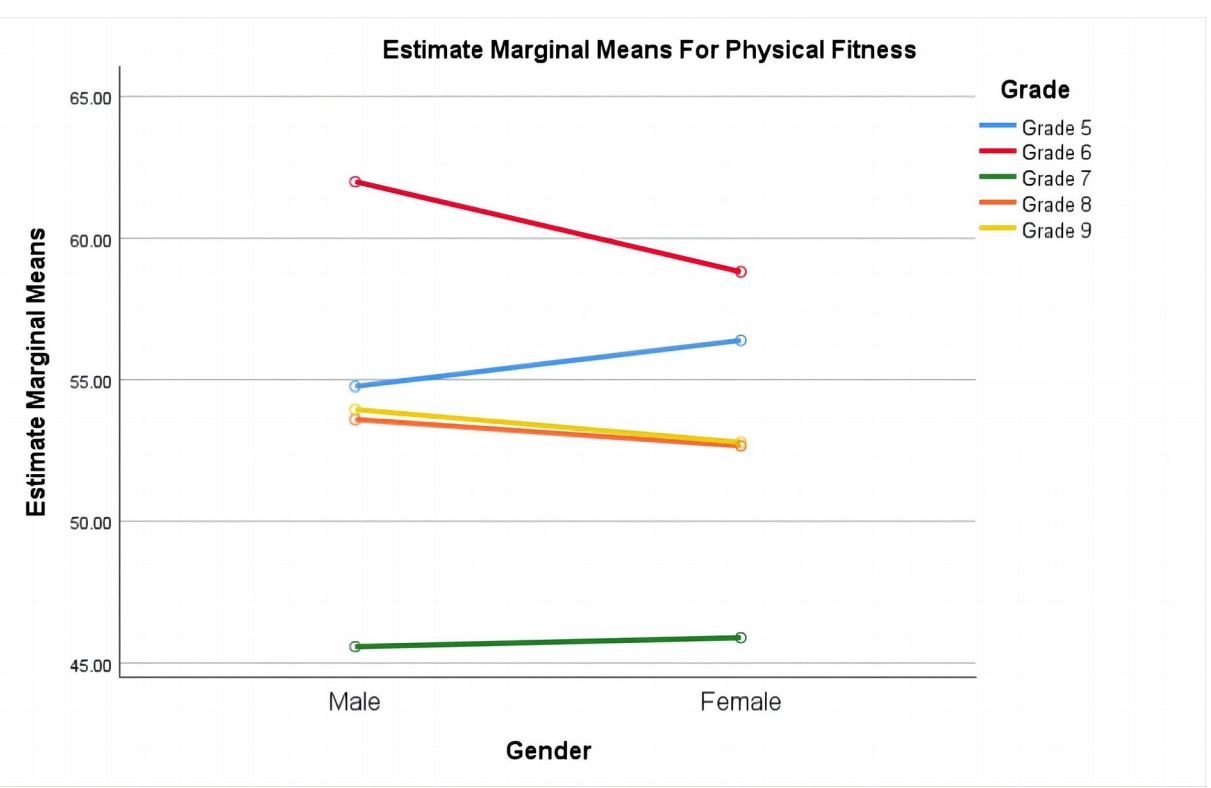

**Fig 2. Gender-grade comparison of physical fitness dimensions.**

interpersonal support (parents, teachers, peers) and adolescent physical literacy. The results, as shown in Table 15, indicate that interpersonal support from parents, teachers, and peers is low to moderately correlated with all dimensions of physical literacy (attitude, sports knowledge, emotion, athletic ability, physical fitness), with correlation coefficients ranging from 0.420 to 0.645.

**4.3.2 The predictive effect of adolescent physical exercise interpersonal support on physical literacy.** While correlation analysis helps explore the relationship between adolescent physical literacy and interpersonal support for physical exercise, it has limitations that may obscure the true relationship. Therefore, all predictors of interpersonal support were entered into a regression model to further examine their impact on adolescent physical literacy.

A multiple regression analysis was conducted with parental support, teacher support, and peer support as independent variables, and gender, grade, and residence type as control variables. The five dimensions of physical literacy were the dependent variables. In the multiple regression analysis, demographic variables (gender, grade, and residence type) were entered in

**Table 11. Multivariate ANOVA overall test of residence type on physical literacy.**

|  | Value | F | Hypothesis *df* | *P* |
|---|---|---|---|---|
| Intercept | .962 | 14981.418 | 5.000 | .000 |
| Residence type | .039 | 11.816 | 10.000 | .000 |

Note: Only Pillai's Trace tests are reported.

**Table 12. One-way ANOVA of residence type on physical literacy.**

| Difference sources | Dependent variable | SQ | DF | MS | F |
|---|---|---|---|---|---|
| Residence type | Attitude | 13148.313 | 2 | 6574.157 | 41.608*** |
| | Sports knowledge | 1737.456 | 2 | 868.728 | 29.412*** |
| | Emotion | 23797.677 | 2 | 11898.839 | 47.221*** |
| | Athletic ability | 3750.203 | 2 | 1875.102 | 42.920*** |
| | Physical fitness | 10179.021 | 2 | 5089.511 | 37.326*** |
| Error | Attitude | 468639.321 | 2966 | 158.004 | |
| | Sports knowledge | 87605.843 | 2966 | 29.537 | |
| | Emotion | 747373.526 | 2966 | 251.980 | |
| | Athletic ability | 129578.024 | 2966 | 43.688 | |
| | Physical fitness | 404422.498 | 2966 | 136.353 | |
| Total | Attitude | 11849161.000 | 2969 | | |
| | Sports knowledge | 1353375.000 | 2969 | | |
| | Emotion | 13555799.000 | 2969 | | |
| | Athletic ability | 2382899.000 | 2969 | | |
| | Physical fitness | 8810159.000 | 2969 | | |

Note

[*]$P<0.05$

[**]$P<0.01$

[***]$P<0.001$.

the first layer of the regression equation, and interpersonal support variables (parents, teachers, peers) were entered in the second layer. The "enter" method was used for regression. The maximum VIF value for collinearity diagnosis was 1.991 < 5, indicating no multicollinearity issues, making it suitable for multiple regression analysis.

The multiple regression results (Table 16) show that parental support, teacher support, and peer support significantly affect the five dimensions of adolescent physical literacy ($P < 0.001$). After controlling for demographic variables, the $\triangle R^2$ for the five dimensions were 0.406, 0.274, 0.369, 0.376, and 0.286, respectively. Peer support made the greatest contribution to predicting all five dimensions of adolescent physical literacy.

## 5.Discussion

### 5.1 Demographic variables and physical literacy

The significant differences observed in the dimensions of physical literacy among adolescents based on gender, grade, and residence type can be attributed to several factors. This section explores these differences in more depth, drawing on the study's quantitative data and theoretical foundations (Social Ecological Model and Self-Determination Theory).

**Table 13. Multiple comparisons of physical literacy among different residence type 1 (M± SD).**

| Category | City | Town | Village |
|---|---|---|---|
| Attitude | 64.22±12.02 | 60.10±12.48 | 59.89±13.40 |
| Sports knowledge | 21.49±5.16 | 19.91±5.23 | 19.97±5.96 |
| Emotion | 68.68±14.89 | 64.17±15.74 | 62.32±17.31 |
| Athletic ability | 28.77±6.20 | 26.69±6.67 | 26.38±7.13 |
| Physical fitness | 55.21±11.04 | 51.91±11.32 | 51.21±12.82 |

**Table 14. Multiple comparisons of physical literacy among different residence type 2(Mean difference).**

| Residence type | | Attitude | Sports knowledge | Emotion | Athletic ability | Physical fitness |
|---|---|---|---|---|---|---|
| City | Town | 4.11*** | 1.57*** | 4.51*** | 2.08*** | 3.31*** |
| City | Village | 4.33*** | 1.51*** | 6.36*** | 2.39*** | 4.00*** |
| Town | Village | .21 | -.06 | 1.85 | .31 | .70 |

Note: ***P<0.001.

## Gender differences

The study found that boys exhibited higher levels of sports knowledge, emotion, and athletic ability compared to girls. This can be attributed to traditional gender roles that associate physical prowess with masculinity, leading to more encouragement and opportunities for boys to engage in physical activities [37]. According to the Social Ecological Model, these differences are influenced by the social environment [38], including family expectations and cultural norms that promote physical activity for boys more than girls [39,40]. Additionally, the Self-Determination Theory suggests that boys might receive more autonomy support from their environment, which fosters higher intrinsic motivation for physical activities [41]. The cultural expectation for boys to be physically active and competitive might also contribute to their higher physical literacy levels [42].

## Grade-Level variations

The study found that younger adolescents (grades 5 and 6) had higher levels of physical literacy compared to older students (grades 7 to 9). This decline in physical literacy with increasing grade level can be attributed to the increased academic pressures and reduced time for physical activities as students progress through school. The Social Ecological Model highlights that school environments and policies significantly impact physical activity behaviors. As academic demands increase, opportunities for physical activity decrease, adversely affecting physical literacy [43]. The Self-Determination Theory further emphasizes that perceived competence and relatedness may decline with higher academic stress and less social support for physical activities [44]. This shift in focus from physical to academic performance as students age likely contributes to the observed decline in physical literacy [45].

**Table 15. Correlation between dimensions of physical literacy and interpersonal support.**

| | Attitude | Sports knowledge | Emotion | Athletic ability | Physical fitness | PAS | TES | PES |
|---|---|---|---|---|---|---|---|---|
| Attitude | 1 | | | | | | | |
| Sports knowledge | .726*** | 1 | | | | | | |
| Emotion | .817*** | .735*** | 1 | | | | | |
| Athletic ability | .775*** | .724*** | .846*** | 1 | | | | |
| Physical fitness | .679*** | .552*** | .693*** | .698*** | 1 | | | |
| PAS | .533*** | .420*** | .503*** | .507*** | .448*** | 1 | | |
| TES | .539*** | .452*** | .521*** | .515*** | .434*** | .577*** | 1 | |
| PES | .584*** | .479*** | .564*** | .570*** | .498*** | .588*** | .645 | 1 |

Note: Gender, grade and residence type are controlled
***P<0.001.

**Table 16. Regression analysis of the predictive effect of interpersonal support on physical literacy dimensions.**

| Category | Attitude | | | | Sports knowledge | | | | Emotion | | | | Athletic ability | | | | Physical fitness | | | |
|---|---|---|---|---|---|---|---|---|---|---|---|---|---|---|---|---|---|---|---|---|
| | B | SE | β | t | B | SE | β | t | B | SE | β | t | B | SE | β | t | B | SE | β | t |
| Gender | -1.308 | .354 | -.051 | -3.690*** | -.772 | .170 | -.070 | -4.535*** | -3.417 | .457 | -.106 | -7.482*** | -.919 | .191 | -.069 | -4.821*** | -1.228 | .362 | -.052 | -3.394** |
| Grade | -.227 | .127 | -.026 | -1.780 | .062 | .061 | .016 | 1.017 | -.702 | .164 | -.063 | -4.275*** | -.188 | .069 | -.040 | -2.744** | -.356 | .130 | -.043 | -2.740** |
| Residence type | -.777 | .216 | -.052 | -3.596*** | -.336 | .104 | -.052 | -3.240** | -1.255 | .278 | -.067 | -4.511*** | -.447 | .116 | -.057 | -3.849*** | -.795 | .220 | -.057 | -3.605*** |
| PAS | .361 | .029 | .229 | 12.509*** | .105 | .014 | .154 | 7.551*** | .396 | .037 | .199 | 10.645*** | .171 | .016 | .207 | 11.035*** | .293 | .029 | .200 | 9.930*** |
| TES | .534 | .052 | .199 | 10.344*** | .222 | .025 | .192 | 8.957*** | .674 | .066 | .198 | 10.138*** | .254 | .028 | .180 | 9.164*** | .308 | .053 | .124 | 5.857*** |
| PES | .837 | .051 | .319 | 16.331*** | .299 | .025 | .265 | 12.135*** | 1.039 | .066 | .314 | 15.740*** | .455 | .028 | .330 | 16.522*** | .727 | .052 | .299 | 13.902*** |
| R² | .431 | | | | .291 | | | | .410 | | | | .405 | | | | .311 | | | |
| R²adj | .430 | | | | .290 | | | | .418 | | | | .404 | | | | .310 | | | |
| △R² | .406 | | | | .274 | | | | .369 | | | | .376 | | | | .286 | | | |
| F | 373.689*** | | | | 203.018*** | | | | 342.624*** | | | | 336.507*** | | | | 222.817*** | | | |

Note
*$P<0.05$
**$P<0.01$
***$P<0.001$.

## Impact of residence type

Adolescents from urban areas exhibited higher levels of physical literacy compared to those from rural areas. Urban environments typically offer better facilities, more organized sports programs, and greater awareness of the benefits of physical activity [9]. This finding is supported by the Social Ecological Model, which underscores the importance of environmental factors, such as access to sports facilities and community support [7,8]. The data suggest that rural adolescents may face barriers such as limited resources and lower parental support for physical activities. The Self-Determination Theory indicates that a lack of autonomy support and relatedness in rural settings may hinder the development of physical literacy [46]. The disparity in access to physical activity opportunities between urban and rural areas can significantly impact adolescents' physical literacy. According to SDT, the fulfillment of autonomy and relatedness needs is crucial for fostering intrinsic motivation and positive behavior. In rural areas, limited resources and opportunities for organized physical activities might reduce adolescents' sense of autonomy and connectedness, thereby affecting their participation in physical activities and overall physical literacy [18,30].

## 5.2 Interpersonal support and physical literacy

Interpersonal support from parents, teachers, and peers significantly influences all dimensions of physical literacy, with peer support showing the highest predictive power. This section delves deeper into these relationships, drawing on the study's data and theoretical frameworks.

**Parental support.** The study's data indicate that parental support positively impacts physical literacy, enhancing children's sports knowledge, emotional well-being, and athletic ability. Parents who encourage and model active behaviors provide autonomy support, fulfilling the basic psychological needs outlined in the Self-Determination Theory. This support fosters intrinsic motivation, leading to higher engagement in physical activities [26,47]. The Social Ecological Model also suggests that parental attitudes and behaviors create a supportive environment that promotes physical literacy. Parental support is crucial in establishing early habits and attitudes towards physical activity [48,49], which can have lasting effects on children's physical literacy.

**Teacher support.** Teachers play a crucial role in developing physical literacy by providing quality physical education and fostering positive attitudes towards physical activity. The study's findings show that teacher support is significantly correlated with students' physical literacy levels. Teachers who create an inclusive and supportive environment enhance students' perceived competence and relatedness, key components of the Self-Determination Theory [20,50]. The Social Ecological Model also emphasizes the impact of school environments and teacher behaviors on students' physical activity levels. Teacher support can help mitigate the negative effects of academic pressures on physical activity by integrating physical literacy into the overall educational experience [51].

**Peer support.** Peer support emerged as the strongest predictor of physical literacy in the study. Adolescents are heavily influenced by their peers, and positive peer interactions can significantly boost physical activity participation. The Self-Determination Theory explains this through the fulfillment of relatedness needs, as supportive peer relationships enhance motivation and engagement in physical activities [52]. The Social Ecological Model also highlights the role of the social environment, including peer groups, in shaping physical activity behaviors. Peer support is particularly influential during adolescence, a period when social acceptance becomes crucial [53]. Adolescents are more likely to engage in and maintain physical activities when they perceive strong peer support and encouragement, making peer influence a vital component of physical literacy development [22,39,54].

## 5.3 Cultural context

Cultural factors significantly influence the dynamics of interpersonal support and physical literacy. This section integrates the cultural context into the analysis, considering the unique aspects of the Chinese setting.

**Parental expectations and support.**   In China, traditional cultural values emphasize academic achievement [47], often at the expense of physical activity [7]. This cultural emphasis can lead to lower levels of parental support for physical activity, particularly if it is perceived as detracting from academic pursuits [48]. This can be traced back to the Confucian value of "scholar above all," where academic success is seen as the primary route to social mobility and family honor. However, when parents recognize the long-term benefits of physical literacy, they may provide more encouragement and resources for their children's participation in physical activities [9]. This shift in parental attitudes is crucial and aligns with the broader recognition of physical health as a foundation for overall well-being and academic success.

**Teacher support in a Confucian context.**   The Confucian educational ethos prevalent in China places teachers in a highly authoritative role, impacting the delivery and perception of physical education [50]. In this context, teachers are seen not only as educators but also as moral guides and disciplinarians. Teachers who adopt a supportive approach can foster greater student engagement in physical activities, thereby enhancing physical literacy [51]. By leveraging their authoritative position, teachers can integrate physical literacy into the academic curriculum and emphasize its importance alongside academic achievements. This dual focus can help students appreciate the value of balancing physical activity with academic pursuits.

**Peer influence and social norms.**   In a culture where academic excellence is highly valued, peer support for physical activity plays a crucial role in shaping adolescents' behaviors [52,53]. The competitive academic environment can sometimes lead to a neglect of physical activities. However, peer groups that value and engage in physical activities can create a supportive environment that encourages participation in sports and exercise [54]. Social norms in schools and communities that promote physical activity can significantly enhance physical literacy [17]. Initiatives that recognize and celebrate physical achievements alongside academic ones can contribute to a more balanced and holistic view of success, encouraging adolescents to pursue both academic and physical excellence.

**The role of modernization and urbanization.**   China's rapid modernization and urbanization have also impacted physical literacy. Urban areas often have better infrastructure and access to sports facilities, which can enhance physical activity levels among adolescents [52]. However, the pressure of urban living and academic competition can also limit the time available for physical activities. In contrast, rural areas may lack adequate facilities, but the traditional lifestyles can sometimes offer more opportunities for physical labor and informal sports [9]. Addressing these disparities requires tailored approaches that consider the unique challenges and advantages of both urban and rural settings.

## 5.4 Limitations

Despite the strengths of this study, there are several limitations that should be acknowledged.

**Absence of control group.**   The absence of a control group limits our ability to directly attribute the observed effects to interpersonal support. Future research should consider including a control group to better isolate the impact of interpersonal support on physical literacy.

**Exclusion of certain regions.**   While our sample is diverse and representative of various regions in China, certain areas such as Tibet, Xinjiang, and Taiwan were excluded. This exclusion may affect the comprehensiveness and representativeness of our findings across the entire country.

**Self-reported data.**   The self-reported nature of the data collection method could introduce response bias, as participants might provide socially desirable answers rather than accurate responses. The potential influence of unaccounted confounding variables cannot be completely ruled out despite our efforts to control for them.

**Generalizability of findings.**   The generalizability of our findings is another important consideration. While our sample is diverse and representative of various regions in China, the results may not be directly applicable to other cultural contexts. The unique cultural, social, and economic factors present in China might influence the dynamics of physical literacy and interpersonal support differently compared to other countries. Therefore, we recommend that future research explore similar relationships in different cultural settings to enhance the generalizability of the results. Additionally, studies that include diverse populations from multiple countries can provide a more comprehensive understanding of the factors influencing physical literacy. This will help in developing more universally applicable strategies to promote physical literacy among adolescents globally.

# 6.Conclusion and enlightenment

## 6.1 Conclusion

1. Boys exhibited higher levels of sports knowledge, emotion, and athletic ability compared to girls. This can be attributed to traditional gender roles and societal expectations that promote physical activity more for boys.

2. Younger adolescents (grades 5 and 6) had higher levels of physical literacy compared to older students (grades 7 to 9), likely due to increasing academic pressures that reduce opportunities for physical activities.

3. Adolescents from urban areas exhibited higher levels of physical literacy compared to those from rural areas, reflecting better access to facilities and organized sports programs in urban environments.

4. The study found that interpersonal support from parents, teachers, and peers significantly influences all dimensions of physical literacy. Peer support, in particular, emerged as the strongest predictor of physical literacy.

## 6.2 Enlightenment

This study provides several important contributions to the field of youth physical literacy and can inform practical applications in various contexts:

1. Educational Policy and Practice: The findings highlight the importance of interpersonal support from parents, teachers, and peers in enhancing adolescents' physical literacy. Policymakers and educational practitioners can leverage these insights to design and implement programs that foster supportive environments both at home and in schools. For instance, integrating physical literacy into the curriculum and promoting teacher training programs focused on providing autonomy support can enhance students' motivation and engagement in physical activities.

2. Parental Involvement Programs: The significant impact of parental support on physical literacy suggests that involving parents in physical education programs could be beneficial. Schools and communities can develop workshops and resources to educate parents on the

importance of physical literacy and ways to support their children's physical activity at home.

3. Peer Support Initiatives: Given the strong influence of peer support, schools can implement peer-led physical activity programs and encourage group sports activities. Creating opportunities for adolescents to engage in physical activities with their peers can enhance their physical literacy and foster a positive attitude towards physical exercise.

4. Targeted Interventions: The study identifies specific demographic variables that influence physical literacy, such as gender, grade, and residence type. These insights can help in developing targeted interventions for groups that may be at a disadvantage. For example, tailored programs for girls or students from rural areas can address specific barriers they face in engaging in physical activities.

5. Future Research Directions: The study opens avenues for further research to explore the underlying mechanisms through which interpersonal support influences physical literacy. Longitudinal studies can provide deeper insights into how these relationships evolve over time and the long-term impact of supportive environments on physical literacy. Exploring these dynamics across different cultural backgrounds and countries will further enhance the understanding and applicability of the findings.

## Supporting information

**S1 Data.**
(RAR)

## Acknowledgments

I would like to thank all participants.

## Author Contributions

**Conceptualization:** Weisong Chen.

**Data curation:** Xuan Wang.

**Formal analysis:** Lin Li.

**Resources:** Lin Li.

**Software:** Bowei Zhou.

**Supervision:** Weisong Chen.

**Writing – original draft:** Xuan Wang.

**Writing – review & editing:** Weisong Chen.

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
