## [Decision Letter · Decision Letter 0]

11 Jun 2024

PONE-D-24-14534Research on relationship between physical literacy and demographic variables and physical exercise interpersonal support of adolescents in ChinaPLOS ONE

Dear Dr. chen,

Thank you for submitting your manuscript to PLOS ONE. After careful consideration, we feel that it has merit but does not fully meet PLOS ONE’s publication criteria as it currently stands. Therefore, we invite you to submit a revised version of the manuscript that addresses the points raised during the review process.

We look forward to receiving your revised manuscript.

Kind regards,

Rogis Baker, Ph.D

Academic Editor

PLOS ONE

Journal Requirements:

4. Please amend the manuscript submission data (via Edit Submission) to include author Lin Li.

Reviewers' comments:

Reviewer's Responses to Questions

**Comments to the Author**

1. Is the manuscript technically sound, and do the data support the conclusions?

Reviewer #1: Yes

Reviewer #2: Yes

Reviewer #3: Yes

Reviewer #4: Yes

2. Has the statistical analysis been performed appropriately and rigorously? 

Reviewer #1: Yes

Reviewer #2: No

Reviewer #3: Yes

Reviewer #4: Yes

3. Have the authors made all data underlying the findings in their manuscript fully available?

Reviewer #1: Yes

Reviewer #2: No

Reviewer #3: Yes

Reviewer #4: Yes

4. Is the manuscript presented in an intelligible fashion and written in standard English?

Reviewer #1: Yes

Reviewer #2: Yes

Reviewer #3: No

Reviewer #4: Yes

5. Review Comments to the Author

Reviewer #1: The manuscript titled "Research on the relationship between physical literacy and demographic variables and physical exercise interpersonal support of adolescents in China" has several concerns in its methods and results sections that need to be addressed for clarity and reliability. Firstly, the sample size justification is missing, which is crucial for validating the study's robustness. The participant selection process lacks detail, raising concerns about potential selection bias. The absence of a control group limits the ability to attribute observed effects directly to interpersonal support. The randomization process, if applied, is not adequately described. Blinding during data collection and analysis is not mentioned, which could introduce bias. The rationale for choosing specific outcome measures is unclear, and the data collection methods lack detailed descriptions, making it difficult to assess the validity and reliability of the data. Statistical methods and the handling of potential confounding variables are not thoroughly explained. There is no discussion on managing missing data, which could impact the results. Additionally, the generalizability of the findings is not addressed, limiting their applicability to broader populations. Lastly, the manuscript does not provide a comprehensive explanation of how cultural factors were integrated into the analysis, which is crucial given the study's focus on the cultural dimension of physical literacy and interpersonal support.

Reviewer #2: I sincerely appreciate the opportunity provided by the journal to review this manuscript. The paper discussed the composition of Physical Literacy among Chinese Youth and the Factors Influencing it. The topic of this study is interesting. The structure of the article is clear and the process is explained well here. However, it could be improved in some detail.

1）The main research of the article is on the factors influencing youth sports literacy, but the realistic contribution part is weak and needs further improvement.

2) For the measurement section, you present each of the breakdown indicators in text form with a few subheadings, kind of like a Postgraduate dissertations format. This section can be expressed through text plus tables, using tables to show the indicators and the weights of the indicators that measure physical literacy in the study, which makes it look more concise.

3) In the data analysis section, the explanation of the research methods used is too simple and it is difficult for scholars who do not have the relevant research background to understand what methods are used in this section and what the methods are useful for, so please explain it in more detail and list out the formulas if you have them.

4) Double check for referencing style and remove errors, proofread the manuscript before next submission as there are typos at many instances.

5) I did not see the relevant data in the supporting information part, please upload it before your next submission.

6) In the explanation of results section, you simply express what is in the table without any further understanding of what causes this one phenomenon, please refine this section further.

Reviewer #3: As a reviewer, I will conduct a comprehensive review of this paper and make the following review comments:

1. Research background and significance: The paper explains the importance of the concept of "Physical Literacy" in the introduction, but it can be further expanded to explain in detail the specific contribution of this study to the realization of the strategic goals of "Healthy China" and "A Leading Sports Nation".

2. Theoretical framework: The paper needs to clearly propose a theoretical framework to support the research. It is recommended that the author introduce more theoretical models on the development of youth physical literacy and clarify the innovations of this study on the basis of existing theories.

3. Research methods: The paper mentions questionnaires and mathematical statistics, but lacks a detailed description of the methods used. It is recommended that the author describe in detail the design process of the questionnaire, the data collection method, and the specific steps of statistical analysis.

4. Sample selection and representativeness: The study used stratified random sampling techniques, but it needs to further explain whether the sample is representative and how to ensure that the sample can reflect the overall situation of Chinese youth physical literacy.

5. Data analysis: The paper provides some descriptive statistics and reliability analysis results, but the detailed description of the data analysis method is insufficient. It is recommended that the authors provide more in-depth analysis of the relationships between variables, such as using path analysis or structural equation modeling to demonstrate the causal relationship between different variables.

6. Discussion and Conclusion: The discussion section needs to explore the implications of the research results in more depth and how these findings are related to existing literature and theory. The conclusion section should clearly point out the main contributions of the study and propose future research directions.

7. Language and writing style: The language of the paper needs further polishing to ensure clarity and professionalism. It is recommended that the author proofread carefully, correct grammatical errors and inaccurate expressions, and consider using professional English editing services for language polishing.

For example: Sports is a powerful weapon to shape and educate people, and the integration of sports and education in the new era takes sports as an important way to "educate people", expecting it to fully embody the essence of life education. Under the guidance of the grand goal of sports power, and the realistic pursuit of deepening the integration of sports and education to promote the healthy development of teenagers, "cultivating people via physical education" is bound to become a new trend in the process of talent training in the future, and the ultimate realization of "cultivating people via physical education" depends on the return of the dominant position of school sports. In this way, the concept of "Physical Literacy" originated from the West just catered to the development needs of the country, the times and individuals.

Problems:

(1)"weapon" may have negative connotations, while "tool" is more neutral and suitable for academic contexts; the first letter of "Sports" is capitalized for consistency.

(2)"integration" was changed to "blending" to reduce repetition, and "utilizes" is more formal than "takes".

(3)To make the sentence more fluent, "expecting" was changed to "with the expectation that" to provide a clear expression of intention.

(4) "realistic pursuit" was changed to "pragmatic pursuit" to emphasize practical actions, and "healthy development" was changed to "healthy growth" to provide a more natural expression.

(5) "depends on" was changed to "hinges on" to provide a stronger expression of dependence, and "dominant position" was changed to "prominent role" to emphasize the core position of school sports.

(6) "In this way" was changed to "Consequently" to emphasize the causal relationship, and "just catered" was changed to "perfectly aligns" to emphasize the perfect fit between concepts and needs.

Suggestion: Read the entire article carefully and improve the language to make it more in line with the writing standards and requirements of SCI journal papers.

Reviewer #4: Dear Authors,

I hope this message finds you well. I am writing to suggest an enhancement to your literature review section. While your current compilation of sources is comprehensive, I believe it would greatly benefit from an increased number of bibliographic references. Including a wider array of sources will not only provide a more robust foundation for your study but also ensure a more thorough exploration of the existing research landscape.

Thank you for considering these suggestions. I am confident that these additions will significantly strengthen the quality and impact of your research.

6. PLOS authors have the option to publish the peer review history of their article (what does this mean?). If published, this will include your full peer review and any attached files.

Reviewer #1: **Yes: **Alex Siu Wing Chan

Reviewer #2: No

Reviewer #3: No

Reviewer #4: No

---

## [Author Response · Author response to Decision Letter 0]

4 Aug 2024

Response to Reviewers

Response to Reviewer 1

Reviewer 1: 

The manuscript titled "Research on the relationship between physical literacy and demographic variables and physical exercise interpersonal support of adolescents in China" has several concerns in its methods and results sections that need to be addressed for clarity and reliability. Firstly, the sample size justification is missing, which is crucial for validating the study's robustness. The participant selection process lacks detail, raising concerns about potential selection bias. The absence of a control group limits the ability to attribute observed effects directly to interpersonal support. The randomization process, if applied, is not adequately described. Blinding during data collection and analysis is not mentioned, which could introduce bias. The rationale for choosing specific outcome measures is unclear, and the data collection methods lack detailed descriptions, making it difficult to assess the validity and reliability of the data. Statistical methods and the handling of potential confounding variables are not thoroughly explained. There is no discussion on managing missing data, which could impact the results. Additionally, the generalizability of the findings is not addressed, limiting their applicability to broader populations. Lastly, the manuscript does not provide a comprehensive explanation of how cultural factors were integrated into the analysis, which is crucial given the study's focus on the cultural dimension of physical literacy and interpersonal support.

Response:

1. Sample Size Justification: We appreciate the reviewer’s suggestion regarding the sample size justification. In the revised manuscript, we have included a detailed sample size justification using both G*Power software and established empirical guidelines within the field of sociology. This information has been added to the Methods section.

2. Participant Selection Process: The participant selection process has been elaborated in the revised manuscript. We have included detailed criteria for participant inclusion and exclusion, and provided information on the selection process to mitigate potential selection bias.

3. Control Group: We acknowledge the absence of a control group in this study. This limitation has been addressed in the Discussion section under the "Limitations" heading. We have discussed the potential impact of this limitation on the study's findings and have suggested directions for future research that could include a control group to better isolate the effects of interpersonal support on physical literacy.

4. Randomization Process: We acknowledge the importance of describing the randomization process in detail. The revised Methods section now provides a comprehensive description of the stratified random sampling technique used to ensure representation across different regions and demographic groups, thereby minimizing potential selection bias.

Participants were selected using a multi-level randomization process:

(1) Stratification by Region: China was divided into five geographical strata based on socio-economic and cultural diversity. Three provinces or cities were selected from each region to ensure balanced representation.

(2) Random Selection of Schools: Within each geographical stratum, schools were selected using a random number generator. The number of schools selected from each region was proportional to the student population in that region.

(3) Random Selection of Students: Within each selected school, students from grades 5 to 9 were chosen using a random number generator. A list of students in each grade was obtained, and a random number generator was used to select the required number of students from these lists.

This detailed randomization process ensured an unbiased allocation of participants, enhancing the study's validity and reliability. These improvements in the description address the concerns regarding the randomization process and its impact on the study findings.

5. Blinding During Data Collection and Analysis: Blinding was implemented during the data collection process to reduce bias. P.E. teachers and head teachers, who administered the questionnaires, were unaware of the specific hypotheses being tested. Additionally, data analysts were blinded to participant identities during analysis. We have added these details to the methods section to address concerns about potential bias.

6. Rationale for Choosing Specific Outcome Measures: We selected our outcome measures based on their relevance and validity in assessing physical literacy and interpersonal support among adolescents. 

The Physical Literacy scale, developed by Xuan Wang (2022), covers five dimensions: emotion, attitude, physical fitness, sports knowledge, and athletic ability. The Physical Exercise Interpersonal Support scale, revised by Liu Jiajing (2021), assesses parental, teacher, and peer support. Both scales have demonstrated high reliability and validity in prior research.

These measures have been adapted and validated within the Chinese context, ensuring cultural relevance. This rationale is detailed in the revised Methods section and has been thoroughly explained in the newly added "2. Theoretical Framework" section.

7. Data Collection Methods: Detailed descriptions of the data collection methods have been added to the Methods section. This includes the procedures followed, the tools used, and the steps taken to ensure the validity and reliability of the data collected.

8. Statistical Methods and Confounding Variables: The statistical methods used in the study and the handling of potential confounding variables have been thoroughly explained in the revised manuscript. We have detailed the statistical techniques employed and the steps taken to control for confounding variables.

9. Managing Missing Data: We have included a discussion on how missing data was managed in the revised manuscript. In the revised Methods section (3.3.5 Handling missing data), we explain that missing data were managed by excluding invalid questionnaires based on predefined criteria, such as incomplete responses and inconsistent answers. This method ensures that only valid responses are included in the analysis, thereby maintaining the quality and integrity of the dataset. 

10. Generalizability of Findings: We have added a section discussing the generalizability of our findings in the revised manuscript. While our sample is diverse and representative of various regions in China, we acknowledge that the findings may not be directly applicable to other cultural contexts. This limitation is discussed in detail in the revised Discussion section. Additionally, the limitations section addresses the exclusion of certain regions such as Xinjiang, Tibet, and Taiwan, which may affect the comprehensiveness of our findings.

We recommend that future research explore similar relationships in different cultural settings to enhance the generalizability of the results. Studies that include diverse populations from multiple countries can provide a more comprehensive understanding of the factors influencing physical literacy. This cross-cultural research will help develop more universally applicable strategies to promote physical literacy among adolescents globally. Moreover, longitudinal studies could offer deeper insights into how the relationships between interpersonal support and physical literacy evolve over time, providing valuable information on the long-term impacts of supportive environments.

11. Integration of Cultural Factors: We have provided a comprehensive explanation of how cultural factors were considered in our discussion. Given the cultural dimension of physical literacy and interpersonal support, we have included a discussion on the cultural relevance of our measures and the interpretation of our findings within the Chinese context. This addition highlights the importance of cultural considerations in understanding physical literacy and interpersonal support among adolescents.

We believe that these revisions address the concerns raised by the reviewer and improve the clarity and reliability of our study. Thank you for your valuable feedback.

Response to Reviewer 2

Reviewer 2:

I sincerely appreciate the opportunity provided by the journal to review this manuscript. The paper discussed the composition of Physical Literacy among Chinese Youth and the Factors Influencing it. The topic of this study is interesting. The structure of the article is clear and the process is explained well here. However, it could be improved in some detail.

1) The main research of the article is on the factors influencing youth sports literacy, but the realistic contribution part is weak and needs further improvement.

Response: Thank you for highlighting this point. In the revised manuscript, we have expanded the Discussion and Conclusion sections to better articulate the practical implications of our findings. Specifically, we have included detailed recommendations for policymakers, educators, and parents on how to leverage interpersonal support to enhance physical literacy among adolescents. These additions aim to provide clearer guidance on how our research can inform real-world practices and interventions.

2) For the measurement section, you present each of the breakdown indicators in text form with a few subheadings, kind of like a Postgraduate dissertations format. This section can be expressed through text plus tables, using tables to show the indicators and the weights of the indicators that measure physical literacy in the study, which makes it look more concise.

Response: We appreciate your suggestion to improve the presentation of the measurement section. In the revised manuscript, we have included tables to summarize the scales that measure physical literacy and interpersonal support. These tables make the section more concise and easier to follow, enhancing the clarity and readability of the manuscript.

3) In the data analysis section, the explanation of the research methods used is too simple and it is difficult for scholars who do not have the relevant research background to understand what methods are used in this section and what the methods are useful for, so please explain it in more detail and list out the formulas if you have them.

Response: We have revised the data analysis section to provide more detailed explanations of the research methods used. We have included descriptions of the statistical techniques, their purposes, and the specific formulas applied in our analysis. This additional detail aims to make the methodology more accessible to scholars from various backgrounds.

4) Double check for referencing style and remove errors, proofread the manuscript before next submission as there are typos at many instances.

Response: We have thoroughly proofread the manuscript to correct any typographical errors and ensure consistency in referencing style. The references have been checked for accuracy and formatted according to the journal's guidelines.

5) I did not see the relevant data in the supporting information part, please upload it before your next submission.

Response: The relevant data has been included in the supporting information section of the revised submission. 

6) In the explanation of results section, you simply express what is in the table without any further understanding of what causes this one phenomenon, please refine this section further.

Response: We acknowledge that the initial Results section lacked detailed analysis. While the Results section itself remains focused on presenting the data, we have significantly expanded the Discussion section to provide a deeper analysis and interpretation of the findings. The revised Discussion section now incorporates theoretical insights and empirical evidence to explain the observed phenomena, offering a comprehensive understanding of the factors contributing to the differences in physical literacy across various demographic variables, as well as the relationship between interpersonal support and physical literacy. This deeper analysis enhances the contextualization of the results and provides a more thorough understanding of the underlying causes.

We believe that these revisions address the concerns raised by the reviewer and improve the clarity and reliability of our study. Thank you for your valuable feedback.

Response to Reviewer 3

Reviewer 3:

As a reviewer, I will conduct a comprehensive review of this paper and make the following review comments:

1. Research background and significance: The paper explains the importance of the concept of "Physical Literacy" in the introduction, but it can be further expanded to explain in detail the specific contribution of this study to the realization of the strategic goals of "Healthy China" and "A Leading Sports Nation".

Response: We appreciate your suggestion to expand on the research background and significance. In the revised manuscript, we have elaborated on the specific contributions of this study to the strategic goals of "Healthy China" and "A Leading Sports Nation". The enhanced introduction now clearly explains how promoting physical literacy among Chinese youth aligns with these national objectives by emphasizing the importance of physical health, lifelong fitness habits, and overall well-being, which are crucial for achieving these strategic goals.

2. Theoretical framework: The paper needs to clearly propose a theoretical framework to support the research. It is recommended that the author introduce more theoretical models on the development of youth physical literacy and clarify the innovations of this study on the basis of existing theories.

Response: We have introduced a more comprehensive theoretical framework in the revised manuscript. This framework incorporates the Social Ecological Model and Self-Determination Theory, which are pivotal for understanding the development of youth physical literacy. Additionally, we have clarified the innovations of this study in relation to existing theories, highlighting how our research extends and applies these models in the Chinese context.

3. Research methods: The paper mentions questionnaires and mathematical statistics, but lacks a detailed description of the methods used. It is recommended that the author describe in detail the design process of the questionnaire, the data collection method, and the specific steps of statistical analysis.

Response: In response to your suggestion, we have detailed the research methods section. The revised manuscript now includes a thorough description of the questionnaire design process, data collection methods, and the specific steps involved in the statistical analysis. These additions provide a clearer understanding of our methodological approach and ensure the robustness and replicability of the study.

4. Sample selection and representativeness: The study used stratified random sampling techniques, but it needs to further explain whether the sample is representative and how to ensure that the sample can reflect the overall situation of Chinese youth physical literacy.

Response: We have expanded the explanation of our sample selection process to address concerns about representativeness. The revised manuscript details the stratified random sampling technique used and discusses how this method ensures that the sample accurately reflects the overall situation of Chinese youth physical literacy. We have also included information on the demographic diversity of the sample to further support its representativeness.

5. Data analysis: The paper provides some descriptive statistics and reliability analysis results, but the detailed description of the data analysis method is insufficient. It is recommended that the authors provide more in-depth analysis of the relationships between variables, such as using path analysis or structural equation modeling to demonstrate the causal relationship between different variables.

Response: We appreciate the suggestion to incorporate path analysis or structural equation modeling to enhance the depth of our data analysis. In this study, we focused on multiple regression analysis to examine the relationships betwee

---

## [Decision Letter · Decision Letter 1]

25 Sep 2024

Research on the relationship between physical literacy and demographic variables and interpersonal support for physical exercise among adolescents in China

PONE-D-24-14534R1

Dear Dr. chen,

We’re pleased to inform you that your manuscript has been judged scientifically suitable for publication and will be formally accepted for publication once it meets all outstanding technical requirements.

Kind regards,

Rogis Baker, Ph.D

Academic Editor

PLOS ONE
---

## [Editor Report · Acceptance letter]

18 Oct 2024

PONE-D-24-14534R1 

PLOS ONE

Dear Dr. Wang, 

I'm pleased to inform you that your manuscript has been deemed suitable for publication in PLOS ONE. Congratulations! Your manuscript is now being handed over to our production team.

Kind regards, 

on behalf of

Dr. Sónia Brito-Costa 

Academic Editor

PLOS ONE